# The KDR Gene rs2071559 and the VEGF Gene rs6921438 May Be Associated with Diabetic Nephropathy in Caucasians with Type 2 Diabetes Mellitus

**DOI:** 10.3390/ijms25179439

**Published:** 2024-08-30

**Authors:** Petra Nussdorfer, David Petrovič, Armin Alibegović, Ines Cilenšek, Danijel Petrovič

**Affiliations:** 1Laboratory for Histology and Genetics of Atherosclerosis and Microvascular Diseases, Faculty of Medicine, University of Ljubljana, Korytkova 2, 1000 Ljubljana, Slovenia; petra.nussdorfer@mf.uni-lj.si (P.N.); ihe@mf.uni-lj.si (D.P.); 2Institute of Histology and Embryology, Faculty of Medicine, University of Ljubljana, Korytkova 2, 1000 Ljubljana, Slovenia; 3Institute of Forensic Medicine, Faculty of Medicine, University of Ljubljana, Korytkova 2, 1000 Ljubljana, Slovenia; armin.alibegovic@mf.uni-lj.si

**Keywords:** diabetic nephropathy, *KDR*, *VEGF*, rs2071559, rs6921438, rs2305948, type 2 diabetes mellitus, VEGF-KDR signalling, association study

## Abstract

The aim of our study was to investigate an association between polymorphisms of either the *VEGF* (vascular endothelial growth factor) gene (rs6921438) or the *KDR* (kinase insert domain receptor) gene (rs2071559, rs2305948) and DN (diabetic nephropathy) in Caucasians with T2DM (type 2 diabetes mellitus). The second aim was to investigate the effect of either the *VEGF* gene (rs6921438) or the *KDR* gene (rs2071559, rs2305948) on the immune expression of either VEGF or KDR in the renal tissues of T2DM subjects (to test the functional significance of tested polymorphisms). The study included 897 Caucasians with T2DM for at least ten years (344 patients with DN and 553 patients without DN). Each subject was genotyped and analyzed for *KDR* (rs1617640, rs2305948) and *VEGF* (rs6921438) polymorphisms. Kidney tissue samples taken from 15 subjects with T2DM (autopsy material) were immunohistochemically stained for the expression of VEGF and KDR. We found that the rs2071559 *KDR* gene was associated with an increased risk of DN. In addition, the GG genotype of the rs6921438 *VEGF* gene had a protective effect. We found a significantly higher numerical area density of VEGF-positive cells in T2DM subjects with the A allele of the rs6921438-*VEGF* compared to the homozygotes for wild type G allele (7.0 ± 2.4/0.1 mm^2^ vs. 1.24 ± 0.5/0.1 mm^2^, respectively; *p* < 0.001). Moreover, a significantly higher numerical area density of KDR-positive cells was found in T2DM subjects with the C allele of rs2071559 (CC + CT genotypes) compared to the homozygotes for wild type T allele (9.7± 3.2/0.1 mm^2^ vs. 1.14 ± 0.5/0.1 mm^2^, respectively; *p* < 0.001) To conclude, our study showed that the presence of the C allele of the rs2071559 *KDR* gene was associated with a higher risk of DN, while the G allele of the rs6921438-*VEGF* conferred protection against DN in Slovenian T2DM subjects.

## 1. Introduction

Type 2 diabetes mellitus (T2DM) is a multifactorial chronic metabolic disease characterized by post-prandial hyperglycemia that causes long-term macrovascular or microvascular complications [1,2,3]. Vascular complications of both the macrovascular and microvascular systems with an increased risk of cardiovascular disease (CVD), diabetic kidney disease (DKD), diabetic retinopathy (DR) and neuropathy are the leading cause of morbidity and mortality in diabetics [4] (Morrish et al., 2001) and represent a huge financial burden [5,6,7,8].

DKD, commonly referred to as diabetic nephropathy (DN), is one of the most important microvascular complications of T2DM [9]. affecting approximately 40% of people with persistent T2DM. It is the leading cause of chronic kidney disease and end-stage renal disease and contributes to increased health problems and deaths from cardiovascular events [10]. 

The classic presentation of DN is characterized by hyperfiltration and persistent albuminuria in the early phases, which is then followed by progressive renal function decline. It is characterized by hypertrophy of the glomeruli, hyperperfusion, thickening of the basement membranes, and a progressive decrease in the glomerular filtration rate (GFR) [1,11,12], which often occurs in association with increased blood pressure and eventually leads to end-stage renal disease [13].

DN is a complex multifactorial disease with a strong genetic component [14]. Many studies have established that genetic susceptibility plays a significant role in the development and progression of DN [15].

Vascular endothelial growth factor (*VEGF*), a key regulator of angiogenesis, has attracted attention as a candidate gene in several diseases, including cardiovascular disease [16], inflammatory diseases [17], diabetes [18], cancer [17], DR [19], and DN [3]. The human *VEGF* gene is located on chromosome 6p21.3 and consists of eight exons interspersed with seven introns [20,21]. Single nucleotide polymorphisms (SNPs) associated or linked to the *VEGF* gene have been correlated with *VEGF* expression and serum levels in T2DN patients [19,22].

In particular, several *VEGF*-related polymorphisms contribute to about half of the variability in circulating *VEGF* levels observed in healthy people. Among these, SNPs within the *VEGF* gene—such as rs833061 and rs699947 (both in the promoter region) as well as rs2010963 (in the 5′ untranslated region [5′ UTR]) and rs3025039 (in the 3′ UTR)—have been most extensively studied in the context of diabetes-related microvascular complications [19,23].

On the one hand, only a few studies have investigated the link between diabetes and diabetic complications and the rs6921438 polymorphism. This genetic variant is located on chromosome 6p21.1, approximately 171 kb downstream from the *VEGF* locus and in close proximity to the *C6orf223* gene, which encodes an uncharacterized protein [24]. The association between these genetic variations, in particular rs6921438, and DN remains uncertain.

The kinase insert domain receptor (KDR), also known as vascular endothelial growth factor receptor 2 (*VEGFR-2* or *Flk-1*), is a *VEGF* receptor. KDR is expressed on the surface of renal endothelial cells, including podocytes and tubular epithelial cells [9,25,26]. It binds vascular endothelial growth factors such as *VEGFA*, *VEGFC,* and *VEGFD* and triggers a cascade of signals that regulates endothelial function and the glomerular filtration barrier in the kidney [13].

The *KDR* gene is located on chromosome 4q11. Polymorphisms in the *KDR* gene, such as rs2071559 and rs2305948, can significantly influence its expression and function. The polymorphism rs2071559 (−604T>C), located in the promoter region of the *KDR* gene at the binding site for the transcriptional factor E2F, has been shown to suppress transcriptional activity and down-regulate the expression of *KDR* [27,28].

The polymorphism rs2305948 (1192G>A), located in exon 7, leads to an amino acid substitution (Val > Ile) at the 297 residues in *KDR* and reduces the binding affinity of *VEGF* to *KDR* [27]. 

These genetic variations in the *KDR* gene have become the focus of research due to their potential associations with various health conditions [16,17,27,28,29,30,31,32,33,34]. However, the specific association of these polymorphisms and DN has not yet been clearly established.

*VEGF* is an endothelial growth factor that promotes angiogenesis and the proliferation and differentiation of endothelial cells. KDR is one of the receptors through which *VEGF* acts. The interaction between *VEGF* and *KDR* affects vascular function, inflammation, and vessel formation, which can impact renal function in conditions like nephropathy [17,35].

The aim of our study was to investigate an association between polymorphisms of either the *VEGF* gene (rs6921438) or the *KDR* gene (rs2071559, rs2305948) and DN in Caucasians with T2DM. The second aim was to investigate the effect of either the *VEGF* gene (rs6921438) or the *KDR* gene (rs2071559, rs2305948) on the expression of either VEGF protein or KDR protein in the renal tissues of T2DM subjects (to test the functional significance of polymorphisms).

## 2. Results

### 2.1. Clinical Characteristics

Table 1 presents the demographic, clinical, and laboratory characteristics of 897 patients, including 553 without DN (control group) and 344 with DN (case group). Significant differences between these groups were observed in several variables, including duration of T2DM, duration of hypertension, SBP, BMI, presence of DR, diabetic neuropathy, DF, levels of S-HbA1c [%], fasting glucose (S-fasting glucose), urea (S-urea), cystatin C (S-cystatin C), creatinine (S-creatinine), triglycerides (S-TGS) and the urine albumin/creatinine ratio (U-albumin/creatinine ratio).

Compared to those without DN, DN patients exhibited a higher prevalence of comorbid chronic diabetic complications such as DR and DF, along with increased BMI and SBP, poorer glycemic control, and extended histories of hypertension and diabetes (Table 1).

### 2.2. Genetical Data

Next, we analyzed the distribution of genotype and allele frequencies of the rs2071559 and rs2305948 polymorphisms of the *KDR* gene in cases and controls (Table 2). For rs2071559, there was a significant difference in the distribution of genotypes and allele frequencies between patients with DN and patients without DN (Table 2).

We also analyzed the distribution of genotype and allele frequencies of the *VEGF* rs6921438 polymorphism. There was a significant difference in the distribution of genotypes and allele frequencies between T2DN patients with DN and patients without DN (Table 3).

In this study, we employed logistic regression analysis adjusting for various factors, including duration of T2DM, hypertension duration, SBP, presence of DKD, diabetic neuropathy, diabetic foot, glycated hemoglobin (HbA1c) levels, fasting glucose (S-fasting glucose), urea, creatinine, cystatin C and urine albumin/creatinine ratio. This approach was undertaken to explore the potential link between selected polymorphisms in the KDR gene (Table 4) and the *VEGF* gene (Table 5) with DN in the Slovenian cohort.

Our risk analysis in the *KDR* gen included three genetic models: dominant, co-dominant, and recessive. The findings, specifically in the dominant and co-dominant models of the rs2071559 polymorphism, revealed a statistically significant association with DN. Conversely, no significant association was observed between the rs2305948 polymorphism and DN, as indicated in Table 4.

In our investigation, we observed that individuals carrying the rs2071559 polymorphism with the homozygous CC genotype exhibited a significantly increased risk of DN. This relationship was statistically significant in both co-dominant and dominant genetic models. Specifically, within the co-dominant model, subjects with the CC genotype were found to have a 1.61-fold increased risk of DN compared to those with the TT genotype (*p* = 0.042). Similarly, individuals with the CT genotype demonstrated a 1.60-fold elevated risk of DN in relation to the TT genotype (*p* = 0.021). Furthermore, our analysis in the dominant model indicated that subjects possessing the C allele (CC + CT genotypes) were associated with a 1.61-fold increased DN risk compared to those with the TT genotype (*p* = 0.013). Conversely, in the recessive model, comparing the CC genotype against the combined CT and TT genotypes, there was no statistically significant association of the CC genotype with DN, as indicated by a p-value of 0.45.

Our study also shows that the rs2305948-*KDR* polymorphism is not associated with DN in T2DM patients (three genetic models: dominant, co-dominant, and recessive) (Table 4).

Moreover, an association between the *VEGF* rs6921438 polymorphism and DN was tested in three genetic models: co-dominant, dominant, and recessive (Table 5). The results in the dominant and recessive models indicate a statistically significant association with DN (Table 5). In individuals with the GG genotype, a 0.51-fold lowered risk for DN (OR =0.51, *p* = 0.004) was found in comparison to subjects carrying the AA genotype. In the dominant model, individuals with the G allele (GG + AG genotypes) had a 0.66-fold lower risk of DN compared to those with the AA genotype (*p* = 0.030). Finally, individuals with the GG genotype had a 0.61-fold lowered risk for DN (OR = 0.61, *p* = 0.009) in comparison to subjects carrying the A allele (recessive model).

### 2.3. Immunohistochemical Data

Regarding the effect of gene polymorphisms on the VEGF/KDR expression, we investigated the effect of either the *VEGF* gene (rs6921438) or the *KDR* gene (rs2071559) on the immune expression of either VEGF or KDR in the renal tissues of T2DM subjects. We found a significantly higher numerical area density of VEGF-positive cells in T2DM subjects with the A allele (AA + AG genotypes) of the rs6921438-VEGF compared to the homozygotes for wild type G allele (7.0 ± 2.4/0.1 mm^2^ vs. 1.24 ± 0.5/0.1 mm^2^, respectively; *p* < 0.001) (Figure 1). Moreover, a significantly higher numerical area density of KDR-positive cells was found in T2DM subjects with the C allele of rs2071559 (CC+CT genotypes) compared to the homozygotes for wild type T allele (9.7 ± 3.2/0.1 mm^2^ vs. 1.14 ± 0.5/0.1 mm^2^, respectively; *p* < 0.001) (Figure 2 and Figure 3). In paraffin sections of renal tissue, KDR protein was detected in glomerular endothelial cells, podocytes, distal tubules, and collecting ducts (Figure 2).

In samples from patients with short-term diabetes, we could also find VEGF immune-expression in glomerular podocytes and distal tubular cells, but the intensity of the immune response was weak (Figure 3). Additionally, some of VEGF positive cells in glomeruli might be endothelial and mesangial cells. However, in samples with advanced changes, VEGF staining was reduced or negative in sclerotic glomeruli but remained intense in tubules (Figure 4 and Figure 5).

## 3. Discussion

Our study demonstrated that the presence of the C allele of the rs2071559 *KDR* gene was associated with a higher risk of DN, while the G allele of the rs6921438-*VEGF* conferred protection against DN in Slovenian T2DM subjects. These findings suggest that genetic variations in these genes play a crucial role in the development of DN among T2DM patients.

To elucidate the influence of genetic variations on the development of DN, we employed linear regression analysis to assess the association between the genotypes of rs2071559 and rs2305948 in the *KDR* gene and the incidence of DN. Our analysis revealed that carriers of the rs2071559 polymorphism have an increased risk of developing DN. This association was evident in both co-dominant and dominant genetic models.

Specifically, CC homozygotes had a 1.61-fold increased risk for DN compared to TT homozygotes, and CT heterozygotes showed a similar risk elevation. These findings underscore the potential significance of the rs2071559 polymorphism in DN pathogenesis. This aligns with previous research that highlighted the role of *KDR* polymorphisms in various micro- and macrovascular complications, including diabetic retinopathy, cardiovascular disease, and CVD [19,33,36]. The results of our study are consistent with those of Kariz and Petrovic [28], who reported an association between the *KDR* rs2071559 polymorphism and myocardial infarction (MI) in Caucasians with T2DM in the Slovenian population. Similarly, Wang et al. [27] observed in Chinese population studies that the rs2071559 polymorphism of *KDR* is associated with an increased risk of coronary heart disease. Merlo and colleagues demonstrated a minor effect of the rs2071559 *KDR* polymorphism on markers of carotid atherosclerosis in subjects with T2DM [16,30,37,38,39]. A study by Yuan et al. investigated the association between the *KDR*-rs2071559 polymorphism and DR in the northern Han Chinese population. The study found that the *KDR* rs2071559 polymorphism was not associated with DR or proliferative DR [34]. A research study from Spain also found no significant link between rs2071559 and age-related macular degeneration [34]. On the other hand, Huang et al. confirmed an association between the rs2071559 *KDR* polymorphism and DR in the Chinese population [40]. While numerous studies have been conducted on the association of polymorphisms in the *KDR* gene with various pathologies, no association with DN has been established despite the potential role of *KDR* in the development of DN [16,34]. Understanding the molecular mechanisms, including genetic factors such as *KDR* polymorphisms, is important for the pathogenesis of diabetes-related vascular disease [2]. The potential pathophysiological mechanism through which the *KDR* polymorphism influences DN development or progression is multifaceted. KDR, also known as VEGFR-2, is a key receptor for VEGF, and its activation plays a critical role in angiogenesis and vascular permeability [17].

VEGF, a potent angiogenic and vascular permeability factor, has been associated with the development of DN in Slovenian patients with T2DM. In the study, logistic regression analysis showed a protective role for the GG genotype of the *VEGF* polymorphism rs6921438 against DN. Individuals with the GG genotype had a significantly lower risk of developing DN compared to those with the AA genotype. This protective effect was consistent across co-dominant, dominant, and recessive models.

Previous studies have shown mixed results regarding the impact of *VEGF* polymorphisms on diabetic complications, making our findings particularly valuable for understanding DN susceptibility. Bonnefond et al. investigated the influence of the rs6921438 polymorphism on the microvascular complications of T2DM, including DR and DN [41]. They found that the G allele of rs6921438, which is linked to higher circulating VEGF levels in the general population, was associated with an increased risk of T2DM and DR in the French population. However, this correlation was not observed in the Danish population. Furthermore, Sajovic et al. found no significant impact of rs6921438 on microvascular complications in T2DM patients [19]. Similarly, Terzić et al. conducted a study in Slovenian patients with T2DM and found that rs6921438 was not associated with DR [24].

Several research groups have previously reported an increase in VEGF and KDR in the retina of T2DM patients [42,43]. Hammes et al. found that the normal retina has little to no expression of VEGF or its receptors, but there is significant synthesis of VEGF and its receptors in the retina of T2DM patients [44]. The study by Aiello et al. and Wang et al. on the T2DM kidney shows that there is a strong correlation between proliferative diabetic retinopathy and DN [45,46]. Furthermore, many of the suspected signaling pathways involved in the development of DR are also thought to play a role in the progression of DN. DR was a good predictor of DN, and PDR predicted DN with high specificity. In patients with T2DM and DN, the severity of DR was associated with glomerular damage [45,46,47].

Increased VEGF levels in diabetic kidneys are linked to glucose stimulation, according to in vitro studies [48]. High glucose levels increase VEGF mRNA expression and protein production in mesangial cells, suggesting a role of VEGF in diabetic kidney disease [48]. The production of VEGF depends on the protein kinase C (PKC) pathway, suggesting that PKC inhibitors may prevent overproduction of VEGF in diabetes. Urinary excretion of VEGF increases with the progression of DN and correlates with serum creatinine levels and proteinuria. In the early stages of DN, upregulation of VEGF is observed in glomerular epithelial cells, whereas in advanced stages, VEGF expression is prominent in the tubular segments, especially in the proximal segment [43,48].

Hakroush et al. highlighted that an increase in VEGF expression is linked with tubulointerstitial fibrosis in proliferative glomerulonephritis and renal tissue remodeling [49,50]. This dysregulated pathway can result in inflammation, endothelial dysfunction, and characteristic DN pathologies [31,32,34,51,52,53,54,55]. 

Chronic hyperglycemia can lead to overexpression of VEGF and abnormal KDR activation [17,56]. Altered KDR expression or activity can disrupt endothelial cell function, production of cytokines, growth factors, and profibrotic mediators [57],, contributing to chronic inflammation, mitochondrial dysfunction, oxidative stress, accumulation of extracellular matrix proteins tubular hypertrophy, glomerular basement membrane thickening, and mesangial expansion. Tubulointerstitial abnormalities can lead to glomerulopathy, fibrosis, and tubular atrophy [52,58,59,60,61,62,63].

Immunohistochemical analysis revealed increased expression of KDR in renal tissues of T2DM patients carrying the rs2071559 CC genotype compared to those with the TT genotype. Similarly, a higher numerical area density of VEGF-positive cells was observed in patients carrying the A allele of rs6921438. These findings support the genetic association data and suggest that these polymorphisms influence the expression of their respective proteins, thereby affecting DN progression.

Thus, in our study, we find that the complex interplay between VEGF signaling and KDR receptor function appears to play a crucial role in DN pathogenesis. VEGF, an important regulator of angiogenesis and vascular permeability, could exacerbate renal damage when overexpressed due to the AA genotype of rs6921438. This effect could be amplified by the rs2071559 polymorphism in *KDR*, which may alter the expression and function of the receptor and lead to impaired endothelial responses. This dual genetic influence underscores the critical role of VEGF-KDR signaling in maintaining renal vascular integrity. Future research should focus on deciphering these signaling pathways in order to develop targeted interventions that mitigate DN risk in genetically predisposed individuals.

Our study has several limitations. Firstly, the sample size, while substantial, may still be insufficient to detect smaller effect sizes or interactions between multiple polymorphisms. Secondly, our analysis was limited to three polymorphisms; other variants in the *VEGF* and *KDR* genes or related pathways might also contribute to DN risk. Thirdly, the study population was restricted to Slovenian patients, and further studies are needed to confirm these findings in other ethnic groups.

## 4. Materials and Methods

### 4.1. Patients

In our cross-sectional case-control study, 897 unrelated Caucasians (Slovenian cohort) with T2DM were enrolled. Investigated patients were recruited from the outpatient clinics of the University Medical Centre Maribor and the General Hospitals in Murska Sobota and Slovenj Gradec. The participants with T2DM were divided into two groups based on their nephrological diagnosis. The first group consisted of 344 patients with DN diagnosed according to the 1999 World Health Organization criteria. The second group, which served as a control group, included 553 subjects with T2DM for at least ten years and without DN. The diagnosis of T2DM was made following the criteria set by the American Diabetes Association. After obtaining informed consent from all participants, we collected their blood samples for analysis. Laboratory tests included measurements of hemoglobin A1c (HbA1c), total cholesterol, high-density lipoprotein (HDL), low-density lipoprotein (LDL), and triglyceride levels. Information on age, gender, blood pressure, duration of T2DM, arterial hypertension (AH) and its duration, body mass index (BMI), smoking habits and the presence of microvascular complications associated with T2DM (specifically, DR and its duration, DN, diabetic foot (DF) as well as the duration of DR and estimated glomerular filtration rate (eGFR) were collected through a questionnaire.

The exclusion criteria for our study were overt nephropathy, active infection, poor glycemic control (glycated hemoglobin (HbA1c) above 10), significant heart failure (New York Heart Association (NYHA) classification II–IV), alcoholism, and the presence of other possible causes of renal disease.

The study protocol was approved by the Slovenian Medical Ethics Committee. The ethics number for our study was 105/12/2011.

### 4.2. Biochemical Analyses

Blood samples from participants were analyzed using standard biochemical methods. Additionally, the albumin/creatinine ratio was determined in three urine samples for each patient according to the diagnostic criteria. The eGFR was estimated using cystatin C and the MDRD study equation.

### 4.3. Genotyping

We extracted genomic DNA from 100 µL of peripheral blood using a QIAamp DNA Blood Mini Kit (Qiagen GmbH, Hilden, Germany). For genotyping of the *KDR* gene (rs2305948 and rs2071559 polymorphisms) and the *VEGF* gene (rs6921438 polymorphism), the StepOne™ (48-well) real-time polymerase chain reaction (PCR) systems from Applied Biosystems by Life Technologies, Foster City, CA, USA, were used in the study. In addition, the TaqMan SNP Genotyping Assay from Applied Biosystems, Foster City, CA, USA, was used. These procedures were performed in accordance with the manufacturer’s instructions. The reaction mixture (5 μL) contained 2.5 μL 2× Master Mix, 0.12 μL 40× Assay Mix, 1.88 μL distilled water Dnase/RNase-free (Gibco, Invitrogen Life Technologies, Waltham, MA, USA), 0.5 μL extracted genomic DNA and oligonucleotide primers labeled with VIC/FAM fluorescent dyes.

### 4.4. Immunohistochemistry

Kidney tissue samples were taken from 15 subjects with T2DM from the Institute of Forensic Medicine (autopsy material). Tissue sections with a thickness of 5 µm were prepared from the kidney samples, fixed in formalin, and embedded in paraffin. The tissue sections were mounted on glass slides and dried. Following standard procedures, the tissues were deparaffinized and dehydrated with graded alcohol solutions. Antigen retrieval was optimized to achieve maximum staining intensity while minimizing background staining. Immunohistochemical staining for KDR and VEGF-positive cells was performed using a Novo Link Max Polymer Detection System (Leica Biosystems Newcastle Ltd., Newcastle upon Tyne, UK). For all immunohistochemical analyses, sections were heated with citrate buffer in a microwave oven (20 min each). Sections were washed with PBS. Endogenous peroxidase and non-specific binding sites were blocked according to the NovoLink kit manufacturer’s instructions. After washing in PBS, the primary antibodies were applied. Specific primary antibodies against KDR (anti-KDR polyclonal antibody, dilution 1:20, Thermo Fischer Scientific, Waltham, MA, USA) and VEGF (anti-VEGF polyclonal antibody, dilution 1:100, Thermo Fischer Scientific, Waltham, MA, USA) were incubated overnight at 4 °C with the tissue sections. Haematoxylin was used for counterstaining. Negative controls were performed by omitting the primary antibodies. Non-tumor kidney tissue was used to establish KDR expression as a positive control, while human lung and skin tissue samples were used to detect VEGF immunoreactivity. The area with KDR- and VEGF-positive cells was delineated, and the numerical area density of positive cells was calculated as the number of positive cells per square millimeter.

### 4.5. Statistical Analysis

In our research, statistical analyses were conducted using IBM SPSS Statistics for Windows, Version 26 (IBM Corp., Armonk, NY, USA). The Hardy-Weinberg equilibrium was assessed via a goodness-of-fit Chi-square test. Differences in discrete variables and genotype distributions between cases and controls were evaluated using the χ^2^ test. The normality of the data distribution was verified using the Shapiro–Wilk test. Continuous variables were compared using the unpaired Student’s t-test for normal distributions and the Mann-Whitney U-test for skewed distributions, with mean ± standard deviation or median values (interquartile range) reported accordingly.

Logistic regression analysis was conducted to explore the association between the rs2305948 and rs2071559 polymorphisms with DN and the association between the rs6921438 polymorphism and DN after adjusting for variables such as duration of T2DM, duration of hypertension, systolic blood pressure (SBP), DR, diabetic neuropathy, DF, glycated hemoglobin (HbA1c), S-fasting glucose, urea, creatinine, cystatin C, and urine albumin/creatinine ratio. This analysis was conducted using co-dominant, dominant, and recessive models. The strength of associations was evaluated using odds ratios (ORs) and 95% confidence intervals (CIs). Associations were considered statistically significant at a *p*-value of less than 0.05.

## 5. Conclusions

In Slovenian patients with T2DM, we demonstrated that the presence of the C allele of the rs2071559 *KDR* gene was associated with a higher risk of DN, while the G allele of the rs6921438-*VEGF* conferred protection against DN in Slovenian T2DM subjects. Furthermore, larger and more diverse cohorts are necessary to validate our findings and understand the broader implications of *VEGF* and *KDR* genetic variations in DN. Our findings suggest a complex interaction between VEGF signaling and KDR receptor function in the development of DN and warrant further studies to elucidate this interplay.

## Figures and Tables

**Figure 1 ijms-25-09439-f001:**
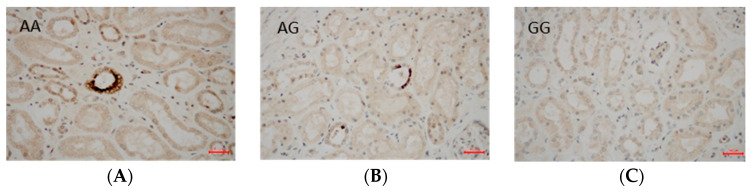
Immunoreaction for VEGF (vascular endothelial growth factor) rs6921438 in tubules of kidney tissue. VEGF-positive cells are stained brown, and VEGF-negative cells are blue. (**A**) immunohistochemically stained of a participant with the AA genotype at 400× magnification; (**B**) immunohistochemically stained of a participant with the AG genotype at 400× magnification; (**C**) immunohistochemically stained of a participant with the GG genotype at 400× magnification.

**Figure 2 ijms-25-09439-f002:**
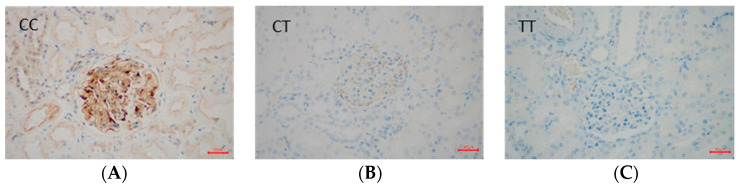
Immunoreaction for KDR (kinase insert domain receptor )of rs2071559 in renal corpuscles of kidney tissue. KDR-positive cells are stained brown, and KDR-negative cells are stained blue. (**A**) immunohistochemical staining of a participant with the CC genotype at 400× magnification; (**B**) immunohistochemical staining of a participant with the CT genotype at 400× magnification; (**C**) immunohistochemical staining of a participant with the TT genotype at 400× magnification.

**Figure 3 ijms-25-09439-f003:**
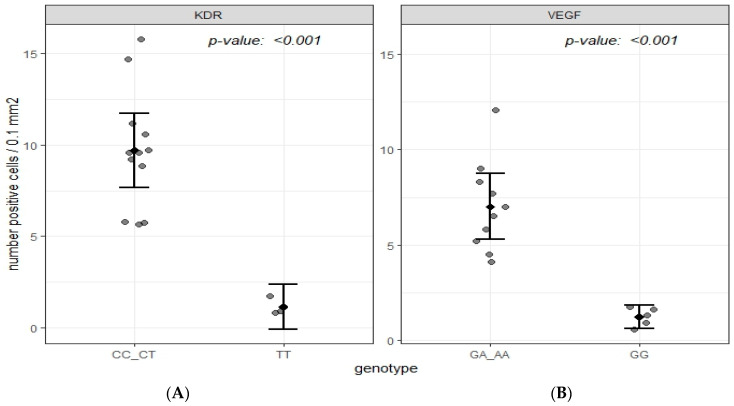
Scatter plot graph comparing the numerical areal density of KDR (kinase insert domain receptor )and VEGF (vascular endothelial growth factor) -positive cells. (**A**) comparing the numerical areal density of KDR-positive cells in patients between CC + CT and TT genotypes. (**B**) comparing the numerical areal density of VEGF-positive cells in patients between GA + GG and GG genotypes.

**Figure 4 ijms-25-09439-f004:**
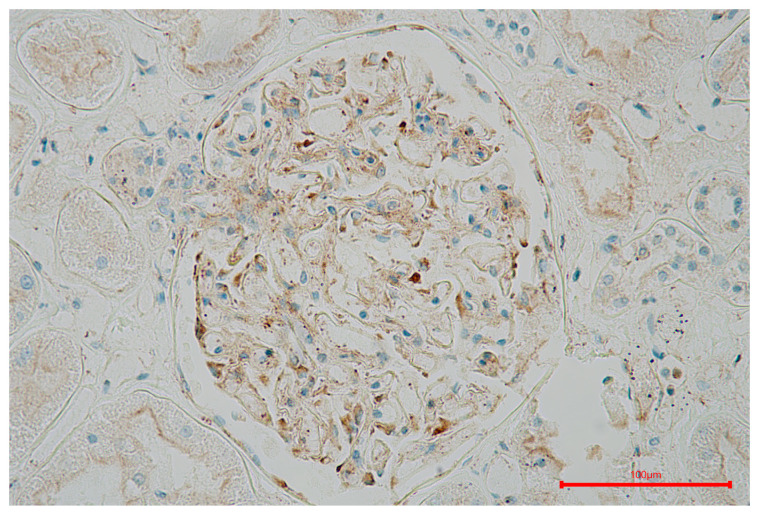
VEGF (vascular endothelial growth factor) is expressed in glomerular podocytes and endothelial and mesangial cells of patients with short-term diabetes.

**Figure 5 ijms-25-09439-f005:**
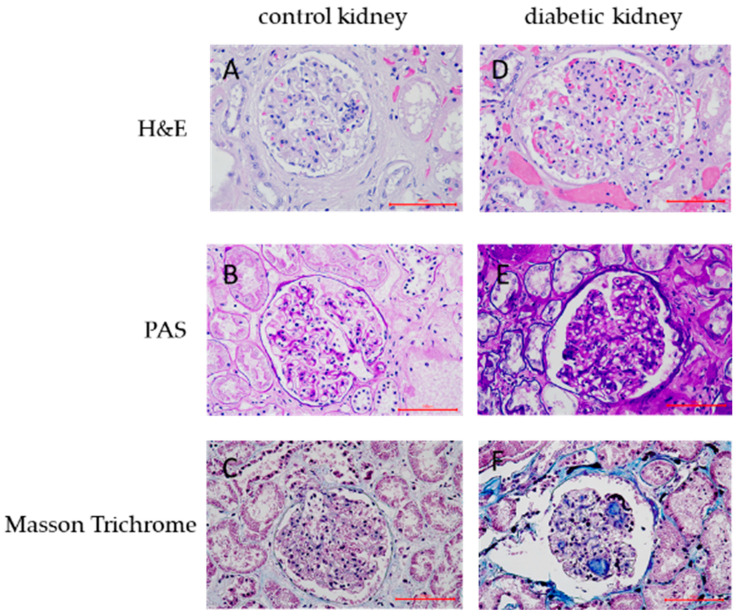
Hematoxylin and eosin (H and E), periodic acid-Schiff (PAS), and Masson trichrome staining of renal tissues in the control and DN (diabetic nephropathy) group. Renal tissues in the control group exhibited normal histological structure (**A**–**C**). Renal tissues in the DN group showed increased mesangial matrix, thickening of the basement membrane, deposition of collagen fibers, and nodular lesions (**D**–**F**) (400× magnification).

**Table 1 ijms-25-09439-t001:** Clinical characteristics and laboratory characteristics of T2DM (type 2 diabetes mellitus), patients with DN (diabetic nephropathy) (cases) and without DN (controls).

	Cases (N = 344)	Controls (N = 553)	*p*-Value
**Sex [M]**	201 (58.4%)	318 (57.5%)	0.78
**Age [years]**	65.35 ± 9.37	65.43 ± 8.63	0.90
**Duration of T2D [years]**	15.69 ± 7.64	13.99 ± 7.00	**<0.001**
**Duration of hypertension [years]**	13.65 ± 8.92	12.18 ± 7.83	**0.020**
**SBP [mmHg]**	155.08 ± 19.4	148.45 ± 19.61	**<0.001**
**DBP [mmHg]**	84.45 ± 11.55	83.09 ± 10.40	0.071
**BMI**	30.91 ± 4.45	30.25 ± 4.52	**0.031**
**Active smokers**	32 (9.3%)	70 (12.7%)	0.12
**CVD**	89 (25.9%)	165 (29.8%)	0.20
**Family history of CVD**			0.14
No	286 (83.1%)	449 (81.2%)	
Yes (before 55 of age)	22 (6.4%)	25 (4.5%)	
Yes (after 55 of age)	36 (10.5%)	79 (14.3%)	
**DR**	149 (43.3%)	115 (20.8%)	**<0.001**
**Duration of DR [years]**	5.00 (4.00–7.00)	5.00 (5.00–8.00)	0.23
**Dneuropathy**	69 (20.1%)	39 (7.1%)	**<0.001**
**DF**	62 (18.0%)	40 (7.2%)	**<0.001**
**S-HbA1c [%] ^1^**	7.93 ± 0.28	7.65 ± 1.14	0.002
**S-fasting glucose [mmol/L]**	8.80 (7.25–10.60)	8.10 (6.80–9.70)	**<0.001**
**S-Hb [g/L]**	138.55 ± 14.17	138.98 ± 12.88	0.69
**S-urea [mmol/L]**	6.40 (5.40–8.20)	6.00 (4.90–7.60)	**<0.001**
**S-creatinine [µmol/L]**	82.00 (68.00–105.00)	77.00 (66.00–91.50)	**<0.001**
** Male sex**	93.00 (75.00–109.00)	84.00 (70.00–98.00)	**<0.001**
** Female sex**	73.00 (57.00–95.25)	70.00 (61.00–82.00)	**0.041**
**eGFR [MDRD equation, mL/min]**	75.00 (60.00–90.00)	76.00 (60.00–90.00)	0.15
** Male sex**	93.00 (75.00–109.00)	84.00 (70.00–98.00)	**<0.001**
** Female sex**	73.00 (57.00–95.25)	70.00 (61.00–82.00)	**0.041**
**S-cystatinC [mg/L]**	0.82 (0.69–1.04)	0.75 (0.65–0.86)	**<0.001**
**S-Totalcholesterol [mmol/L]**	4.40 (3.80–5.23)	4.40 (3.90–5.20)	0.41
**S-HDL [mmol/L]**	1.20 (1.00–1.40)	1.20 (1.00–1.40)	0.83
**S-LDL [mmol/L]**	2.50 (2.00–3.10)	2.50 (2.00–3.10)	0.45
**S-TGS [mmol/L]**	1.60 (1.10–2.50)	1.50 (1.00–2.20)	**0.023**
**U-albumin/creatinine ratio [g/moL], sample no. 1**	8.14 (3.80–23.60)	0.99 (0.60–1.58)	**<0.001**
**U-albumin/creatinine ratio [g/moL], sample no. 2**	8.26 (3.70–25.10)	1.02 (0.68–1.70)	**<0.001**
**U-albumin/creatinine ratio [g/moL], sample no. 3**	7.98 (3.82–22.82)	1.02 (0.68–1.73)	**<0.001**

Statistically significant values are written in bold. Legend: DN: diabetic nephropathy; T2DM: Type 2 diabetes mellitus; SBP: Systolic blood pressure; DBP: Diastolic blood pressure; BMI: Body mass index; CVD: Cardiovascular disease; DR: Diabetic retinopathy; DF: Diabetic foot; S-HbA1c [%] ^1^: Glycated hemoglobin; Hb: serum hemoglobin; eGFR: estimated glomerular filtration rate; S-TGS: triglyceride; HDL: High-density lipoprotein; LDL: Low-density lipoprotein.

**Table 2 ijms-25-09439-t002:** Distribution of rs2071559 and rs2305948 genotypes and alleles in the case group (patients with T2DM (type 2 diabetes mellitus) and DN (diabetic nephropathy)) and control group (patients with T2DM without DN).

	Cases (N = 344)	Controls (N = 553)	*p*-Value
** *KDR* ** **_rs2071559**			
** CC**	88 (25.6%)	122 (22.1%)	**0.034**
** CT**	182 (52.9%)	269 (48.6%)	
** TT**	74 (21.5%)	162 (29.3%)	
** *KDR* ** **_rs2305948**			
** TT**	2 (0.6%)	11 (2.0%)	0.16
** CT**	73 (21.2%)	103 (18.6%)	
** CC**	269 (78.2%)	439 (79.4%)	
**HWE**			
** *KDR* ** **_rs2071559**	0.2667	0.6048	
** *KDR* ** **_rs2305948**	0.2104	0.0950	
**ALLELES**			
** *KDR* ** **_rs2071559**			
** C (MAF)**	358 (52.0%)	513 (46.4%)	**0.020**
** T**	330 (48.0%)	593 (53.6%)	
** *KDR* ** **_rs2305948**			
** T (MAF)**	77 (11.2%)	125 (11.3%)	0.94
** C**	611 (88.8%)	981 (88.7%)	
**DOMINANT**			
** *KDR* ** **_rs2071559**			
** CC + CT**	270 (78.5%)	391 (70.7%)	**0.010**
** TT**	74 (21.5%)	162 (29.3%)	
** *KDR* ** **_rs2305948**			
** TT + CT**	75 (21.8%)	114 (20.6%)	0.67
** CC**	269 (78.2%)	439 (79.4%)	
**RECESSIVE**			
** *KDR* ** **_rs2071559**			
** CC**	88 (25.6%)	122 (22.1%)	0.23
** CT + TT**	256 (74.4%)	431 (77.9%)	
** *KDR* ** **_rs2305948**			
** TT**	2 (0.6%)	11 (2.0%)	0.086
** CT + CC**	342 (99.4%)	542 (98.0%)	
** CT + CC**	342 (99.4%)	542 (98.0%)	

Statistically significant values are written in bold. Legend: DN: diabetic nephropathy; T2DM: Type 2 diabetes mellitus; HWE: Hardy-Weinberg equilibrium.

**Table 3 ijms-25-09439-t003:** Distribution ofrs6921438 genotypes and alleles in the case group (patients with T2DM (type 2 diabetes mellitus) and DN (diabetic nephropathy)) and control group (patients with T2DM without DN).

*VEGF*_rs6921438	Cases (N = 344)	Controls (N = 553)	*p*-Value
** GG**	82 (23.8%)	180 (32.5%)	** *0.007* **
** AG**	176 (51.2%)	271 (49.0%)	
** AA**	86 (25.0%)	102 (18.4%)	
**ALLELES**			
** G (MAF)**	340 (49.4%)	631 (57.1%)	** *0.002* **
** A**	348 (50.6%)	475 (42.9%)	
**HWE (*p*-value)**	0.2667	0.6048	
**DOMINANT**			
** GG + AG**	258 (75.0%)	451 (81.6%)	** *0.019* **
** AA**	86 (25.0%)	102 (18.4%)	
**RECESSIVE**			
** GG**	82 (23.8%)	180 (32.5%)	** *0.005* **
** AG + AA**	262 (76.2%)	373 (67.5%)	

**Table 4 ijms-25-09439-t004:** Logistic regression analysis adjusted for different variables according to genetic models.

*KDR*_rs20715599	Count	OR (95% CI)	*p*-Value for OR
**co-dominant**			
CC vs. TT	88/122 vs. 74/162	1.61 (1.02–2.56)	0.042
CT vs. TT	182/269 vs. 74/162	1.60 (1.08–2.39)	0.021
**dominant**			
[CC + CT] vs. TT	270/391 vs. 74/162	1.61 (1.11–2.35)	0.013
**Recessive**			
CC vs. [CT + TT]	88/122 vs. 256/431	1.16 (0.79–1.68)	0.45
** *KDR* ** **_rs2305948**	**count**	**OR (95% CI)**	***p*-value for OR**
**co-dominant**			
TT vs. CC	2/11 vs. 269/439	0.21 (0.01–1.38)	0.17
CT vs. CC	73/103 vs. 269/439	1.53 (0.80–2.93)	0.20
**dominant**			
[TT + CT] vs. CC	75/114 vs. 269/439	1.27 (0.69–2.35)	0.44
**recessive**			
TT vs. [CT + CC]	2/11 vs. 342/542	0.21 (0.01–1.32)	0.16

Adjusted for: duration of T2DM (type 2 diabetes mellitus), duration of hypertension, SBP (systolic blood pressure), DR (diabetic retinopathy), diabetic neuropathy, DF (diabetic foot), HbA1c (gycated hemoglobin), S-fasting glucose, urea, creatinine, cystatin C, and urine albumin/creatinine ratio.

**Table 5 ijms-25-09439-t005:** Logistic regression analysis adjusted for different variables according to genetic models.

*VEGF* _rs6921438	Count	OR (95% CI)	*p*-Value for OR
**co-dominant**			
GG vs. AA	82/180 vs. 86/102	0.51 (0.32–0.8)	0.004
AG vs. AA	176/271 vs. 86/102	0.76 (0.5–1.13)	0.17
**dominant**			
[GG + AG] vs. AA	258/451 vs. 86/102	0.66 (0.45–0.96)	0.030
**Recessive**			
GG vs. [AG + AA]	82/180 vs. 262/373	0.61 (0.43–0.88)	0.009

Adjusted for: duration of T2DM (type 2 diabetes mellitus), duration of hypertension, SBP (systolic blood pressure), DR (diabetic retinopathy), diabetic neuropathy, DF (diabetic foot), HbA1c (gycated hemoglobin), S-fasting glucose, urea, creatinine, cystatin C, and urine albumin/creatinine ratio.

## Data Availability

The data presented in this study are available at the request of the corresponding author due to sensitive information (patients’ clinical data).

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
