# Peer review of "The KDR Gene rs2071559 and the VEGF Gene rs6921438 May Be Associated with Diabetic Nephropathy in Caucasians with Type 2 Diabetes Mellitus"

_ijms, 2024, doi:10.3390/ijms25179439_

Round 1

Reviewer 1 Report

Comments and Suggestions for Authors

The study of Nussdorfer and colleagues investigates possible suscepitibility factors contributing to the development of diabetic kidney disease, a major health care burden worldwide. The study aim is excellent, study design is appropriate but several concerns have to be addressed and many errors corrected.

1) In the abstract the first sentence does not make sense in present form "affect renal function, like diabetic nephropathy" - DN is not a renal function (?).

2) Line 30 of the abstract is a repetition of line 28 - please revise.

3) In the introduction, line 80, the sentence "KDR is expressed on the surface of endothelial cells, including podocytes and tubular epithelial cells" is nonsense, as the 2 latter cell types are not endothelial cells (!)

4) Line 84, "KDR gene, encoding, is located" some words are missing here (copy/paste issue?)

5) In methods, line 111: first sentence makes no sense in present form, please rephrase!

6) Lines 135-137: please specify how were the biochemical parameters measured (using which reagents, what kind of laboratory analyzers (manufacturer, type, etc)).

7) There is no description of kidney histology from the biopsy samples. However, basic histology analysis is the first step after a biopsy, so please include a descriptive table with detailed histology scoring of the used samples (tubular dilatation / atrophy / fibrosis score / glomerulosclerosis score / vascular damage score).

8) Immunohistochemistry results should be correlated to histology. Also, histology could be correlated to UACR or eGFR.

9) In section 2.4, the selection criteria of the kidney biopsies need to be described. 

10) Line 158: Please describe how was antigen retrieval performed! Also, specify the blocking step and the used antibody diluent.

11) Line 164: H&E can not be used for counterstain as it would dissolve completely the immunostaining. Please correct!

12) Line 168-171 is the same as line 166-168, clear redundancies.

13) In the immunohistochemical methods description, you should describe how was the staining evaluated: magnification, type of microscope and digital camera, type of software for quantification.

14) The results section could be improved a lot by converting Table 2 and Table 3 into a graph.

15) Table 4 and 5 title says "regression adjusted for different variables" - please describe in detail for which variables and how was it adjusted.

16) Together with the renal immunostainings, representative renal histology pictures should be shown as well (same biopsies as for the immunostaining photographs) to enable correlating the two analysis results.

17) In figures 1 and 2, unfortunately there is no analysis result shown, that was mentioned in the methods section. Quantitation of immunostainings with appropriate statistical evaluation is required, and correlation analysis to fibrosis scores from basic histology (PAS stain or Masson's).

18) Fig 2 legend is incorrect: A1-A3 should be B1-B3 according to the figure.

19) Were glomeruli negative for VEGF staining? Only tubules are shown in the pictures. Podocytes are the main source of VEGF regulating glomerular endothelial cell function and growth.

20) In figure 2, please specify which glomerular cells stain positive for KDR? A double staining is preferable with mesenchymal, endothelial and podocyte markers.

21) If the VEGF receptor appeared only in the glomeruli but VEGF only in the tubules, how could this represent a functional signaling, how would you associate tubular VEGF excretion to glomerular VEGF receptor function?

22) In the discussion, line 436-437 is not supported by the results (!).

23) The discussion section could be shortened by 20% at least.

Comments on the Quality of English Language

Some copy/paste errors and preseverations need to be corrected, as well as few sentences without proper meaning in the present form. Otherwise only minor grammar errors were detected.

Author Response

Reviewer 1

The study of Nussdorfer and colleagues investigates possible suscepitibility factors contributing to the development of diabetic kidney disease, a major health care burden worldwide. The study aim is excellent, study design is appropriate but several concerns have to be addressed and many errors corrected.

1) In the abstract the first sentence does not make sense in present form "affect renal function, like diabetic nephropathy" - DN is not a renal function (?).

Reply: Thank you for you comment. We revised the first sentence accordingly.

The interaction between vascular endothelial growth factor (VEGF) and kinase insert domain receptor (KDR) influences vascular function, inflammation and vascularisation, which may affect renal function.

2) Line 30 of the abstract is a repetition of line 28 - please revise.

Reply: Thank you for you comment. We revised the abstract accordingly.

 Alterations in receptor expression and function can impair endothelial responses. Our study associates the rs2071559 KDR and rs6921438 VEGF polymorphisms with the risk of DN in Slovenian subjects with T2DM, focusing and confirming the role of the VEGF-KDR signaling pathway.

3) In the introduction, line 80, the sentence "KDR is expressed on the surface of endothelial cells, including podocytes and tubular epithelial cells" is nonsense, as the 2 latter cell types are not endothelial cells (!) 

Reply: Thank you for you comment. We revised the paper accordingly, and we correct the sentence.

KDR is expressed on the surface of renal endothelial cells, podocytes and tubular epithelial cells.

4) Line 84, "KDR gene, encoding, is located" some words are missing here (copy/paste issue?)

Reply: Thank you for you comment. We revised the paper accordingly, and we correct the sentence.

The KDR gene is located on chromosome 4q11.

5) In methods, line 111: first sentence makes no sense in present form, please rephrase!

Reply: Thank you for you comment. We revised the paper accordingly, and we correct the sentence.

now line 388: In our cross-sectional case-control study 897 unrelated Caucasians (Slovenian cohort) with T2DM were enrolled.  

6) Lines 135-137: please specify how were the biochemical parameters measured (using which reagents, what kind of laboratory analyzers (manufacturer, type, etc)).

Reply: Thank you for your comment. We have revised the article accordingly and added the missing information

 now line 411 - 415 : Plasma  glucose,  S-Hb,  glycated  hemoglobin  (HbA1c),  urea,  creatinine, cystatin C, total cholesterol, LDLs, HDLs, and tri-glycerides  (TGs)  were  determined  by  standard  biochemical  methods. Albumin/creatinine ratio (ACR) was determined in three urine samples for each patient  according to diagnostic criteria.

All biochemical investigations were done on a fully automated analyzer at the University Medical Centre Maribor and General Hospitals, Murska Sobota and Slovenj Gradec.

7) There is no description of kidney histology from the biopsy samples. However, basic histology analysis is the first step after a biopsy, so please include a descriptive table with detailed histology scoring of the used samples (tubular dilatation / atrophy / fibrosis score / glomerulosclerosis score / vascular damage score). 

Reply: Thank you for your comment. We have revised the article regarding the histological changes in diabetic kidney disease.

  1. We wanted to empasize that we used autopsy cases from the Institue for forensic medicine, and not the biopsy material from subjects with diabetic nephropathy. In our group of subjects (15 autopsy cases), there were different stages of renal involvment, from minimal changes to mostly sclerotic glomeruli in few of them. There were some clinical data available for most of them (for example serum urea and kreatinin levels, history of diseases), but not all data in all of them. All of them came to the Institute for forensic medicine due to sudden death, most of them due to sudden cardiac death (occlusion of coronary artery, ventricular fibbrilation, asystole…). For all subjects we had crucial clinical and laboratory data.
  2. Regarding the histology - the second aim was to investigate the effect of either the VEGF gene (rs6921438) or the KDR gene (rs2071559, rs2305948) on the expression of either VEGF gene or KDR gene in the renal tissues of T2DM subjects (to test the functional significance of polymorphisms). Therefore, we wanted to test the functional significance of polymorphisms (i.e. to find potential association between gene polymoprhism (VEGF, KDR) and expression of tested gene (VEGF, KDR), and not to evaluate the histological changes of diabetic kindey disease).
  3. However, since our autopsy cases of sudden death with DM had different duration of type 2 DM, we could observe variours degree of histological changes from minor to almost completely scklerotic glomuruli

8) Immunohistochemistry results should be correlated to histology. Also, histology could be correlated to UACR or eGFR.

Reply: Thank you for your comment. We have revised the article regarding the histological changes in diabetic kidney disease.

  1. We wanted to empasize that we used autopsy cases from the Institue for forensic medicine, and not the biopsy material from subjects with diabetic nephropathy. In our group of subjects (15 autopsy cases), there were different stages of renal involvment, from minimal changes to mostly sclerotic glomeruli in few of them. There were some clinical data available for most of them (for example serum urea and kreatinin levels, history of diseases), but not all data in all of them. All of them came to the Institute for forensic medicine due to sudden death, most of them due to sudden cardiac death (occlusion of coronary artery, ventricular fibbrilation, asystole…).
  2. For all subjects we had crucial clinical and laboratory data regarding the second aim. We had data such as eGFR, serum urea and creatinin from some, but not all of them.

9) In section 2.4 the selection criteria of the kidney biopsies need to be described. 

Reply: Thank you for your comment. We have revised the article accordingly and added the missing information.

However, we must state again that there were no biopsy material, since we used autopsy cases from the Institue for forensic medicine. For the purpose of our study (second aim of the study), we wanted to test the functional significance of polymorphisms (i.e. to find potential association between gene polymoprhism (VEGF, KDR) and expression of tested gene (VEGF, KDR), and not to evaluate the histological changes of diabetic kindey disease) (see also answer 7, paragraph b).

4.4.): Kidney tissue samples were obtained at autopsy from 15 subjects with T2DM.

10) Line 158: Please describe how was antigen retrieval performed! Also, specify the blocking step and the used antibody diluent.

Reply: Thank you for your comment. We have revised the article accordingly and added the missing information

now line 435-445: For all immunohistochemical analyses, sections were heated with citrate buffer in a microwave oven (20 minutes each). Sections were washed with PBS. Endogenous peroxidase and non-specific binding sites were blocked according to the NovoLink kit manufacturer's instructions. After washing in PBS, the primary antibodies were applied. Specific primary antibodies against KDR (anti-KDR polyclonal antibody, dilution 1:20, Thermo Fischer Scientific) and VEGF (anti-VEGF polyclonal antibody, dilution 1:100, Thermo Fischer Scientific) were incubated overnight at 4°C with the tissue sections

11) Line 164: H&E can not be used for counterstain as it would dissolve completely the immunostaining. Please correct!

Reply: Thank you for your comment. We revised the paper accordingly, and we correct the sentence.

now line 445: Haematoxylin was used for counterstaining.

12) Line 168-171 is the same as line 166-168, clear redundancies.

Reply: Thank you for your comment. We revised the paper accordingly, and we omit the sentence:

now line 447-448: Non-tumorous kidney tissue served as a positive control for KDR expression, while human lung and skin tissue samples were used to detect VEGF immunoreactivity.

13) In the immunohistochemical methods description, you should describe how was the staining evaluated: magnification, type of microscope and digital camera, type of software for quantification.

Reply: Thank you for you comment. We revised the paper accordingly, and added the missing information.

After staining, the slides were independently read and evaluated under light microscope Axio Scope 2 (Zeiss Group, Germany) by two researchers blinded to the genotyping results. For each subject, the immunoprofile was evaluated by averaging two tissue-slides. The immunoreactive cells were counted manually at 400× magnification. The counts were normalized to number of cells per unit of kidney tissue area (number of cells/mm2).

14) The results section could be improved a lot by converting Table 2 and Table 3 into a graph.

Reply: Thank you for you comment. It is an interesting idea. Here is one major problem if we do this: we would have in Table 2 8 graphs, and in table 3 additional 3 graph. Due to that fact, we think that it is more rationale to stay with this format.

15) Table 4 and 5 title says "regression adjusted for different variables" - please describe in detail for which variables and how was it adjusted.

Reply: Thank you for you comment. We used logistic regression analysis.

Text in the methods section: Logistic regression analysis was conducted to explore the association between the rs2305948 and rs2071559 polymorphisms with DN and the association between the rs6921438 polymorphism and DN after adjusting for variables such as duration of T2DM, duration of hypertension, systolic blood pressure (SBP), DR, diabetic neuropathy, DF, glycated haemoglobin (HbA1c), S-fasting glucose, urea, creatinine, cystatin C, and urine albumin/creatinine ratio.

Regarding the Table 4 – adjustment was made for: duration of T2DM, duration of hypertension, systolic blood pressure, diabetic retinopathy, diabetic neuropathy, diabetic foot, HbA1c, S-fasting glucose, urea, creatinine, cystatin C, and urine albumin/creatinine ratio.

Regarding the Table 5 – adjustment was made for: duration of T2DM, duration of hypertension, systolic blood pressure, diabetic retinopathy, diabetic neuropathy, diabetic foot, HbA1c, S-fasting glucose, urea, creatinine, cystatin C, and urine albumin/creatinine ratio.

16) Together with the renal immunostainings, representative renal histology pictures should be shown as well (same biopsies as for the immunostaining photographs) to enable correlating the two analysis results.

Reply: Thank you for you comment. We have previously performed other staining (such as PAS, Masson tricrome) to see what kind of renal tissue we had from the Institute of forensic medicine (see below). However, since this was not our objective, we did not put them in the paper. We demonstrated different histological changes. We found however, for example, especially Masson tricrome very convinient for the evaluation of the degree of sclerosis. However, as stated previously, this was not the aim of our current study.

Regarding the second aim was to investigate the effect of either the VEGF gene (rs6921438) or the KDR gene (rs2071559, rs2305948) on the expression of either VEGF gene or KDR gene in the renal tissues of T2DM subjects (to test the functional significance of polymorphisms).   Therefore, we wanted to test the functional significance of polymorphisms (i.e. to find potential association between gene polymoprhism (VEGF, KDR) and expression of tested gene (VEGF, KDR), and not to evaluate the histological changes of diabetic kindey disease).

Fig. Glomerulus from T2DM patient with nodular glomerulosclerosis (Masson trichrome stain), 400x magnification

Fig. Glomerulus from T2DM patient with diffuse mesangial expansion (Periodic Acid Schiff’s (PAS) stain), 400x magnification.

17) In figures 1 and 2, unfortunately there is no analysis result shown, that was mentioned in the methods section. Quantitation of immunostainings with appropriate statistical evaluation is required, and correlation analysis to fibrosis scores from basic histology (PAS stain or Masson's).

Reply: Thank you for you comment. We have previously performed other staining (such as PAS, Masson tricrome) to see what kind of renal tissue we had from the Institute of forensic medicine (see below). However, since this was not our objective, we did not put them in the paper. We demonstrated different histological changes. We found however, for example, especially Masson tricrome very convinient for the evaluation of the degree of sclerosis. However, as stated previously, this was not the aim of our current study.

Fig. Glomerulus from T2DM patient with nodular glomerulosclerosis (Masson trichrome stain), 400x magnification

18) Fig 2 legend is incorrect: A1-A3 should be B1-B3 according to the figure.

Reply: Thank you for you comment. We revised the paper accordingly, and we correct the legend.

Figure 2. Immunoreaction for KDR of rs2071559 in renal corpuscles of kidney tissue. KDR-positive cells are stained brown, KDR-negative cells are stained blue. (B1) immunohisto-chemical staining of a participant with the CC genotype at 400× magnification; (B2) immuno-histochemical staining of a participant with the CT genotype at 400× magnification; (B3) im-munohistochemical staining of a participant with the TT genotype at 400× magnification.

19) Were glomeruli negative for VEGF staining? Only tubules are shown in the pictures. Podocytes are the main source of VEGF regulating glomerular endothelial cell function and growth.

Reply: Thank you for you comment. In glomeruli we did not find any expression of VEGF(napredovalo stanje).

In our study, like other research groups (Cha et al.2000, Cooper et al.1999), we found that in biopsies with mild changes in diabetic nephropathy, VEGF was increased in glomerular podocytes and distal tubule cells. In biopsies with advanced changes, VEGF staining was reduced or negative in sclerotic glomeruli but remained intense in the tubules. In the early stages of DN, upregulation of VEGF is observed in glomerular epithelial cells, whereas in advanced stages, VEGF expression is prominent in the tubular segments, especially in the proximal segment (Williams et al.; 2024 Nardi et al.2020).

20) In figure 2, please specify which glomerular cells stain positive for KDR? A double staining is preferable with mesenchymal, endothelial and podocyte markers.

Reply: Thank you for you comment. We revised the paper accordingly, and added the missing information. We immunolabel glomerular endothelial cells (black arrows) and podocytes (red arrows). We tried double staining, but we don't have a standardised protocols and the results were unusable.  We agree with the reviewer that we should do further staining with other markers, but first we need to standardise the staining protocols.

21) If the VEGF receptor appeared only in the glomeruli but VEGF only in the tubules, how could this represent a functional signaling, how would you associate tubular VEGF excretion to glomerular VEGF receptor function?

Reply: Thank you for you comment. We revised the paper accordingly.  We have slides of subjects with diabetes (autopsy material Institute of Forensic Medicine) with different degree of renal dysfuncion. In those with progressive disease, higher percentage of sclerotic glomeruli are present in comparison with early stages.

22) In the discussion, line 436-437 is not supported by the results (!).

Reply: Thank you for you comment. We revised the paragraph regarding the second aim of the study in the final part of the Result section – about the effect of either the VEGF gene (rs6921438) or the KDR gene (rs2071559) on the expression of either VEGF gene or KDR gene in the renal tissues of T2DM subjects.

Last paragraph of the Result section:

Regarding the effect of gene polymorphisms on the VEGF/KDR expression, we investigated the effect of either the VEGF gene (rs6921438) or the KDR gene (rs2071559) on the expression of either VEGF gene or KDR gene in the renal tissues of T2DM subjects. We found a significantly higher numerical area density of VEGF-positive cells in T2DM subjects with the A allele (AA+AG genotypes) of the rs6921438-VEGF compared to the homozygotes for wild type G allele (7,0± 2.4/0,1mm2 vs. 1,24 ± 0,5/0,1mm2, respectively; p < 0.001) (Figure 1). Moreover, a significantly higher numerical area density of KDR-positive cells was found in T2DM subjects with the C allele of rs2071559 (CC+CT genotypes) compared to the homozygotes for wild type T allele (9,7± 3,2/0,1mm2 vs. 1,14 ± 0,5/0,1mm2, respectively; p < 0.001) (Figure 2). In paraffin sections of renal tissue, KDR protein was detected in glomerular endothelial cells, podocytes, distal tubules and collecting ducts (Fig.2).

23) The discussion section could be shortened by 20% at least.

Reply: Thank you for you comment. We revised the paper accordingly, and we shortened the discussion section.

Last 2 paragraphs were substantially shortened.

Some copy/paste errors and preseverations need to be corrected, as well as few sentences without proper meaning in the present form. Otherwise only minor grammar errors were detected.

Submission Date

27 June 2024

Date of this review

05 Jul 2024 20:59:09

Reviewer 2 Report

Comments and Suggestions for Authors

The submitted manuscript explores the associations between KDR gene (rs2305948 and rs2071559) and VEGF gene (rs 6921438) polymorphisms and the presence of diabetic kidney disease. It also explores the tissue expression of KDR and VEGF in kidney biopsies of patients with type 2 diabetes. The manuscript is overall well written and organized and my comments are mostly minor:

- I would modify the title deleting "in Slovenian patients with type 2 diabetes" and adding the translational approach (i.e. histological evaluation). 

- I would organize the results section in sub-paragraphs

- Angiogenesis plays a crucial role in diabetic-associated complications. In this scenario, it would be important to understand if SNPs rs2071559 and rs6921438 lead to a different physiology and pathophysiology of VEGFR2 and VEGF, respectively. E.g. angiogenesis is regulated by CD93 (also called C1qR1) and has been associated with diabetic complications such as DN (PMID: 31549396 PMID: 37371490). Understanding if SNPs are associated with different interactions with CD93 or other angiogenic players would be helpful to develop novel treatments.  

Author Response

The submitted manuscript explores the associations between KDR gene (rs2305948 and rs2071559) and VEGF gene (rs 6921438) polymorphisms and the presence of diabetic kidney disease. It also explores the tissue expression of KDR and VEGF in kidney biopsies of patients with type 2 diabetes. The manuscript is overall well written and organized and my comments are mostly minor:

- I would modify the title deleting "in Slovenian patients with type 2 diabetes" and adding the translational approach (i.e. histological evaluation).

Reply: Thank you for you suggestion. We revised the title.

  1. New title: The KDR gene rs2071559 and the VEGF gene rs 6921438 may be associated with diabetic nephropathy in Caucasians with type 2 diabetes mellitus
  2. Regarding the histology - the second aim was to investigate the effect of either the VEGF gene (rs6921438) or the KDR gene (rs2071559, rs2305948) on the expression of either VEGF gene or KDR gene in the renal tissues of T2DM subjects (to test the functional significance of polymorphisms). Therefore, we wanted to test the functional significance of polymorphisms (i.e. to find potential association between gene polymoprhism (VEGF, KDR) and expression of tested gene (VEGF, KDR), and not to evaluate the histological changes of diabetic kindey disease).

- I would organize the results section in sub-paragraphs

Reply: Thank you for you suggestion. We organized the results section in sub-paragraphs.

- Angiogenesis plays a crucial role in diabetic-associated complications. In this scenario, it would be important to understand if SNPs rs2071559 and rs6921438 lead to a different physiology and pathophysiology of VEGFR2 and VEGF, respectively. E.g. angiogenesis is regulated by CD93 (also called C1qR1) and has been associated with diabetic complications such as DN (PMID: 31549396;   PMID: 37371490). Understanding if SNPs are associated with different interactions with CD93 or other angiogenic players would be helpful to develop novel treatments.  

Reply: Thank you for you suggestions and comments. In our study we sought for the association between tested polymorphisms and DN. We reported that the presence of the C allele of the rs2071559 KDR gene was associated with higher risk of DN, while the G allele of the rs6921438-VEGF conferred protection against DN in Slovenian T2DM subjects.

We are aware of the importance of CD93 in angiongenesis in various disorders, and as a potential new biomarker of DN. However, so far there are not many reports about the role of CD93 in DN. Additionally, there are three gene polymorphisms of CD93 gene available (rs2749817, rs2749812 and rs3746731), however we have not tested them yet.

Role of CD93 in Health and Disease.

Tossetta G, Piani F, Borghi C, Marzioni D. Cells. 2023 Jul 4;12(13):1778. doi: 10.3390/cells12131778. PMID: 37443812 Free PMC article. Review.

3

Diagnostic and Prognostic Role of CD93 in Cardiovascular Disease: A Systematic Review.

Piani F, Tossetta G, Cara-Fuentes G, Agnoletti D, Marzioni D, Borghi C. Biomolecules. 2023 May 30;13(6):910. doi: 10.3390/biom13060910.

Lee M, Park HS, Choi MY, Kim HZ, Moon SJ, Ha JY, Choi A, Park YW, Park JS, Shin EC, Ahn CW, Kang S. Significance of Soluble CD93 in Type 2 Diabetes as a Biomarker for Diabetic Nephropathy: Integrated Results from Human and Rodent Studies.  J Clin Med. 2020 May 8;9(5):1394. doi: 10.3390/jcm9051394.

Glomerular endothelial cell-podocyte stresses and crosstalk in structurally normal kidney transplants.

Menon R, Otto EA, Berthier CC, Nair V, Farkash EA, Hodgin JB, Yang Y, Luo J, Woodside KJ, Zamani H, Norman SP, Wiggins RC, Kretzler M, Naik AS. Kidney Int. 2022 Apr;101(4):779-792. doi: 10.1016/j.kint.2021.11.031. Epub 2021 Dec 22. PMID: 34952098

Targeted Proteomics Reveals Functional Targets for Early Diabetes Susceptibility in Young Adults.

Shah RV, Zhong J, Massier L, Tanriverdi K, Hwang SJ, Haessler J, Nayor M, Zhao S, Perry AS, Wilkins JT, Shadyab AH, Manson JE, Martin L, Levy D, Kooperberg C, Freedman JE, Rydén M, Murthy VL. Circ Genom Precis Med. 2024 Feb;17(1):e004192. doi: 10.1161/CIRCGEN.123.004192. Epub 2024 Feb 7. PMID: 38323454

Round 2

Reviewer 1 Report

Comments and Suggestions for Authors

The manuscript has been improved, but still contains errors to be corrected, and an important previous question remained unanswered.

1) Line 112-113 and line 175: VEGF and KDR gene expression was not investigated, but protein expression (immunostaining)!? Quantitative gene expression analysis is missing. Please clarify.

2) For Figure 1 and 2, quantification of the immunostainings should be added as a graph showing the results of each immunostaining quantification (prefereably a scatter plot) along with appropriate statistical evaluation.

3) Unfortunately Comment 21 from the first revew was not answered by the authors. So the main question is still the same, if the VEGF receptor (KDR) appeared only in the glomeruli but VEGF only in the tubules, how could this represent a functional signaling, how would you associate tubular VEGF excretion to glomerular VEGF receptor function? This needs appropriate interpretation!

4) If PAS and Masson stainings were performed, please include these as representative histology pictures in a separate figure. I disagree that histology evaluation is of minor importance in this particular study. Histology and renal function deterioration has a major impact on VEGF and KDR production, unrelated to genotypes. Also, different genotypes could have an association with progression rate, therefore with eGFR, ACR and histology. This is a crucial information that somehow the authors do not want to share in detail, unfortunately, substantially reducing the value of the manuscript.

Comments on the Quality of English Language

English is overall fine, some typos and minor grammar errors need correction.

Author Response

ODGOVORI

The manuscript has been improved, but still contains errors to be corrected, and an important previous question remained unanswered.

1) Line 112-113 and line 175: VEGF and KDR gene expression was not investigated, but protein expression (immunostaining)!? Quantitative gene expression analysis is missing. Please clarify.

Reply: Thank you for your comment. We have revised the paper accordingly and corrected the sentences. We used VEGF/KDR immune-expression instead.

Modified lines 112-113: The second aim was to investigate the effect of either the VEGF gene (rs6921438) or the KDR gene (rs2071559, rs2305948) on the immune-expression of either VEGF or KDR in the renal tissues of T2DM subjects (to test the functional significance of tested polymorphisms).

Modified line 175: Regarding the effect of gene polymorphisms on the VEGF/KDR expression, we investigated the effect of either the VEGF gene (rs6921438) or the KDR gene (rs2071559) on the immune-expression of either VEGF or KDR in the renal tissues of T2DM subjects.

2) For Figure 1 and 2, quantification of the immunostainings should be added as a graph showing the results of each immunostaining quantification (preferably a scatter plot) along with appropriate statistical evaluation.

Reply: Thank you for your comment. We have revised the article accordingly and added the missing information. We added a scatter plot for appropriate genotypes (Figure 3).

Figure 3. Scatter plot graph comparing the numerical areal density of KDR and VEGF- positive cells. (A) comparing the numerical areal density of KDR-positive cells in patients between CC+CT and TT genotype. (B) comparing the numerical areal density of VEGF-positive cells in patients between GA+GG and GG genotype.

3) Unfortunately Comment 21 from the first review was not answered by the authors. So the main question is still the same, if the VEGF receptor (KDR) appeared only in the glomeruli but VEGF only in the tubules, how could this represent a functional signaling, how would you associate tubular VEGF excretion to glomerular VEGF receptor function? This needs appropriate interpretation!

Reply: Thank you for your comment. We have corrected the article accordingly. We have added slides of samples with short-term diabetes showing different degrees of renal dysfunction. A higher percentage of sclerotic glomeruli is present in progressive disease compared to early stages.

In samples from patients with short-term diabetes, we can also find VEGF expressed in glomerular podocytes and distal tubular cells, as shown below (figure 3). However, in samples with advanced changes, VEGF staining was reduced or negative in sclerotic glomeruli but remained intense in tubules (figure 4) .

We identified only a small number of podocytes that were VEGF-immunopositive in patients with short-term diabetes, but the intensity of the immune response was weak and limited. Additionally, some of VEGF positive cells in glomeruli may be endothelial and mesangial cells. Similarly, VEFF immune-expression was demonstrated in few tubular cells (distal tubules and collecting ducts).

In our study, the majority of diabetic glomeruli did not show a positive response to anti-VEGF antibodies, whereas in few we found immune-reactivity to VEG, in contrast to the data of other authors (1,2,3). Other authors also showed the absence of a positive immune response in glomerular cells in normal and tumour-adjacent renal parenchyma (4). Moreover, in those slides immune-reactivity was present not only in glomeruli, but also in tubular cells in subjects with diabetes.

1.Brown LF, Berse B, Tognazzi K, Manseau EJ, Van de Water L, Senger DR, Dvorak HF, Rosen S. Vascular permeability factor mRNA and protein expression in human kidney. Kidney Int. 1992 Dec;42(6):1457-61. doi: 10.1038/ki.1992.441. PMID: 1474780.

  1. Dvorak HF, Nagy JA, Feng D, Brown LF, Dvorak AM. Vascular permeability factor/vascular endothelial growth factor and the significance of microvascular hyperpermeability in angiogenesis. Curr Top Microbiol Immunol. 1999;237:97-132. doi: 10.1007/978-3-642-59953-8_6. PMID: 9893348.

3.Burt LE, Forbes MS, Thornhill BA, Kiley SC, Chevalier RL. Renal vascular endothelial growth factor in neonatal obstructive nephropathy. I. Endogenous VEGF. Am J Physiol Renal Physiol. 2007 Jan;292(1):F158-67. doi: 10.1152/ajprenal.00293.2005. Epub 2006 Jun 20. PMID: 16788140.

4.Baderca F, Lighezan R, Dema A, Alexa A, Raica M. Immunohistochemical expression of VEGF in normal human renal parenchyma. Rom J Morphol Embryol. 2006;47(4):315-22. PMID: 17392976.

  1. Rioux-Leclercq N, Fergelot P, Zerrouki S, Leray E, Jouan F, Bellaud P, Epstein JI, Patard JJ. Plasma level and tissue expression of vascular endothelial growth factor in renal cell carcinoma: a prospective study of 50 cases. Hum Pathol. 2007 Oct;38(10):1489-95. doi: 10.1016/j.humpath.2007.02.014. Epub 2007 Jun 26. PMID: 17597181

Figure 4. VEGF expressed in glomerular podocytes, endothelial and mesangial cells of patients with short-term diabetes

Vascular endothelial growth factor (VEGF) is a paracrine secreted product of podocytes under physiological conditions and is involved in endothelial cell homeostasis [1]; however, pathological conditions such as DN result in paracrine and autocrine hypersecretion from podocytes [2]. This results in the typical renal pathology of glomerular hypertrophy, glomerular basement membrane (GBM) thickening, mesangial matrix enlargement, hilar membrane loss, and podocyte loss [2]. The etiology of DN is thought to be decreased production of endothelial nitric oxide synthase (eNOS) in vascular endothelial cells and increased production of VEGF in podocytes (separation of VEGF and endothelial monoxide) due to endothelial cell damage from reactive oxygen species and the resulting advanced glycation end products. in the hyperglycemic environment caused by diabetes mellitus (DM) [3, 4]. In the early stage of DN, overproduction of VEGF causes abnormal renal pathology, but as the disease progresses, podocyte necrosis and glomerulosclerosis prevent VEGF production, leading to further endothelial cell damage [5].

  1. Sison K, Eremina V, Baelde H, Min W, Hirashima M, Fantus IG, et al.. Glomerular structure and function require paracrine, not autocrine, VEGF-VEGFR-2 signaling. J Am Soc Nephrol. 2010;21(10):1691–701. 10.1681/ASN.2010030295. [PMC free article] [PubMed] [CrossRef] [Google Scholar]
  2. Veron D, Reidy KJ, Bertuccio C, Teichman J, Villegas G, Jimenez J, et al.. Overexpression of VEGF-A in podocytes of adult mice causes glomerular disease. Kidney Int. 2010;77(11):989–99. 10.1038/ki.2010.64. [PubMed] [CrossRef] [Google Scholar]
  3. Nakagawa T, Sato W, Glushakova O, Heinig M, Clarke T, Campbell-Thompson M, et al.. Diabetic endothelial nitric oxide synthase knockout mice develop advanced diabetic nephropathy. J Am Soc Nephrol. 2007;18(2):539–50. 10.1681/ASN.2006050459. [PubMed] [CrossRef] [Google Scholar]
  4. Nakagawa T. Uncoupling of the VEGF-endothelial nitric oxide axis in diabetic nephropathy: an explanation for the paradoxical effects of VEGF in renal disease. Am J Physiol Ren Physiol. 2007;292(6):F1665–72. 10.1152/ajprenal.00495.2006. [PubMed] [CrossRef] [Google Scholar]
  5. Hernández-Arteaga K, Soto-Abraham V, Pérez-Navarro M, de León-Garza B, Rodríguez-Matías A, Olvera-Soto MG, et al.. Thrombotic microangiopathy in patients with diabetic nephropathy is associated with low VEGF expression and end-stage renal disease. Clin Nephrol. 2018;89(6):429–37. 10.5414/CN109240. [PubMed] [CrossRef] [Google Scholar]

4) If PAS and Masson stainings were performed, please include these as representative histology pictures in a separate figure. I disagree that histology evaluation is of minor importance in this particular study. Histology and renal function deterioration has a major impact on VEGF and KDR production, unrelated to genotypes. Also, different genotypes could have an association with progression rate, therefore with eGFR, ACR and histology. This is a crucial information that somehow the authors do not want to share in detail, unfortunately, substantially reducing the value of the manuscript.

Reply: Thank you for your comment.

  1. We agree with the reviewer that histological changes in diabetic kidney disease have major impact on renal function deteriorations, and that this has a major impact on VEGF and KDR production, unrelated to genotypes. For example, in case of completely sclerotic glomeruli there is no immune-expression of VEGF or KDR in podocytes. Due to that fact we added figures 4 and 5- example of short-term diabetes.
  2. There are however there are only few reports on VEGF immune-expression in normal kidney. Baderca et al. did not report immunoreaction to VEGF in most cells of renal corpuscle . In fact only in one case of normal case (without diabetes) immunoreaction to VEGF in some isolated podocytes was found. At the same time there was positive reaction in tubules and collecting ducts. Intense immune-expression might be related to most important VEGF function (i.e. effect on vascular permeability).
  3. Again, I would like to stress that we wanted to test the functional significance of polymorphisms (i.e. to find potential association between gene polymorphism (VEGF, KDR) and expression of tested gene (VEGF, KDR), and not to evaluate the histological changes of diabetic kidney disease) with regard to different clinical settings in our group of subjects from the Institute of Forensic Medicine. Subjects from the Institute of Forensic Medicine generally have less clinically and laboratory significant data.
  4. Hopefully we will get in future some cases (slides) from the Institute of pathology, however this is not expected in near future. Regarding the slides from the Institute of pathology (biopsy specimens) better slides are expected with more preserved tissue which is especially relevant for analyses of tubules and glomeruli. Subjects who have renal biopsy due to diagnostic evaluation have more clinical and laboratory relevant data. But at the same time, we are aware of the fact that renal biopsy is not performed for diagnostic evaluation of diabetics, since the diagnosis is made according to clinical data (history of diabetes, ultrasound, laboratory: urine and blood).

4.Baderca F, Lighezan R, Dema A, Alexa A, Raica M. Immunohistochemical expression of VEGF in normal human renal parenchyma. Rom J Morphol Embryol. 2006;47(4):315-22. PMID: 17392976.

We have revised the paper accordingly and attached representative histological images

Figure 5. Hematoxylin and eosin (H&E), periodic acid-Schiff (PAS) and Masson trichrome staining of renal tissues in the control and DN group. Renal tissues in the control group exhibited normal histological structure (A-C). Renal tissues in the DN group showing increased mesangial matrix, thickening of the basement membrane, deposition of collagen fibers, and nodular lesions (D-F) (400× magnification).

Round 3

Reviewer 1 Report

Comments and Suggestions for Authors

The authors now improved the paper significantly. My questions have been answered.